# Making a science out of preanalytics: An analytical method to determine optimal tissue fixation in real-time

**Daniel R. Bauer** [1]*, **Torsten Leibold**[2], **David R. Chafin**[1]

1 Roche Tissue Diagnostics (Ventana Medical Systems, Inc.), Tucson, Arizona, United States of America,
2 Raytheon Missiles & Defense, Tucson, Arizona, United States of America

* daniel.bauer.db2@roche.com

**Data Availability Statement:** All relevant data are within the manuscript and its Supporting information files.

**Funding:** The author(s) received no specific funding for this work.

## Abstract

Modern histopathology is built on the cornerstone principle of tissue fixation, however there are currently no analytical methods of detecting fixation and as a result, in clinical practice fixation is highly variable and a persistent source of error. We have previously shown that immersion in cold formalin followed by heated formalin is beneficial for preservation of histo-morphology and have combined two-temperature fixation with ultra-sensitive acoustic monitoring technology that can actively detect formalin diffusing into a tissue. Here we expand on our previous work by developing a predictive statistical model to determine when a tissue is properly diffused based on the real-time acoustic signal. We trained the model based on the morphology and characteristic diffusion curves of 30 tonsil cores. To test our model, a set of 87 different tonsil samples were fixed with four different protocols: dynamic fixation according to our predictive algorithm (C/H:Dynamic, N = 18), gold-standard 24 hour room temperature (RT:24hr, N = 24), 6 hours in cold formalin followed by 1 hour in heated formalin (C/H:6 +1, N = 21), and 2 hours in cold formalin followed by 1 hour in heated formalin (C/H:2+1, N = 24). Digital pathology analysis revealed that the C/H:Dynamic samples had FOXP3 staining that was spatially uniform and statistically equivalent to RT:24hr and C/H:6+1 fixation protocols. For comparison, the intentionally underfixed C/H:2+1 samples had significantly suppressed FOXP3 staining ($p<0.002$). Furthermore, our dynamic fixation protocol produced bcl-2 staining concordant with standard fixation techniques. The dynamically fixed samples were on average only submerged in cold formalin for 4.2 hours, representing a significant workflow improvement. We have successfully demonstrated a first-of-its-kind analytical method to assess the quality of fixation in real-time and have confirmed its performance with quantitative analysis of downstream staining. This innovative technology could be used to ensure high-quality and standardized staining as part of an expedited and fully documented preanalytical workflow.

**Competing interests:** DB, TL and DC were full-time employees of Roche Tissue Diagnostics during this work and received funding for experiments from Roche Tissue Diagnostics. There are no products in development or marketed products to declare. Roche Tissue Diagnostics has filed patents on the TOF technology and associated algorithms. DB, TL and DC explicitly acknowledge this does not alter our adherence to PLOS ONE policies on sharing data and materials." (as detailed online in our guide for authors http://journals.plos.org/plosone/s/competing-interests).

# Introduction

Clinical tissue processing techniques "fix" tissues with crosslinking agents that shut down metabolism within cells and preserve crisp and clear cellular morphology. The most prevalent fixative is 10% neutral buffered formalin (NBF) which is an aqueous solution of formaldehyde in a buffer and has been used for over a century [1]. Currently, proper fixation protocols are empirically determined by examining the histologically stained tissue for proper morphological features. There have been some published guidelines from the College of American Pathologists to provide better tissue handling standards, including suggested times of fixation for most tissues in room temperature formalin [2]. The result is a mixed bag of adequate and poor morphology depending on the operator, institution, tissue type, and biomarker of interest. Furthermore, by the time the morphology is interrogated, it is too late to improve the quality of the tissue so proper fixation the first time is critically important.

Tissue fixation is a time consuming process usually taking several hours to days depending on the type and size of the tissue. As pressures mount to decrease the turnaround time for patient care, rapid fixation protocols are being introduced. One such technology already being employed is to raise the temperature of the fixative to increase the crosslinking rate [3–21]. While this is practically effective, the use of increased fixative temperature has led to many reports of unsatisfactory tissue morphology based on hematoxylin and eosin (H&E) stain and variability in other molecular analysis, such as routine immunohistochemical (IHC) stains [22–24]. Biologically, the use of heated fixative serves to crosslink proteins on the outside of the tissue sample while compromising the structure of proteins in the middle where sufficient fixative has not penetrated. A more promising rapid method that produces superior tissue fixation uses 10% NBF in two-temperature zones (cold+hot) [25, 26]. The cold step allows proper diffusion of formaldehyde into the tissue interior followed by a short heated phase that rapidly forms formaldehyde-based crosslinks.

Presently there are no established methods of actively monitoring either the diffusion of formaldehyde into tissues or the actual formation of crosslinks. Inadequately diffused tissues will only crosslink and fix where the fixative has penetrated, forming an outer ring of well-fixed tissue. Inadequate fixation is a leading cause of reported errors in anatomical pathology laboratories [27–32]. Because current tissue fixation protocols lack real-time monitoring there is no way to guarantee sufficient formaldehyde concentrations in the tissue or conversely to know if a sample has become overfixed, which also has detrimental effects on stain quality [10, 33]. In short, current fixation techniques offer no quality assurance or tracking capability to tissue processing laboratories. This means when samples are improperly fixed expensive rework is required, if another sample is capable of being procured again at all. Although several methods for statically detecting diffusion exist including optical, ultrasound, and MRI, these detection mechanisms have not been implemented for tissue processing to better preserve cancer indicators [34–38]. Some researchers have soaked tissues in radioactive formaldehyde and measured fluid penetration after exposure to photographic film as a measure of diffusion rates [39]. However, very little radioactivity is actually incorporated into the tissue and long exposure times led to fuzzy and unreliable results. Others have used ultrasound monitoring to look at crosslinking by comparing samples of unfixed to fixed tissues [40–45]. Ultimately, neither of these techniques allows for monitoring in real-time when changes could be implemented to guarantee excellent tissue fixation and proper functional staining.

Therefore, to ensure sufficient formaldehyde was present throughout a tissue to guarantee proper staining, we developed a dynamic method to optimize tissue fixation using real-time detection of formalin diffusion. We have previously described an automated system that can detect penetration of a fixative into a raw biological tissue using acoustic time-of-flight (TOF)

technology that exploits the discrete sound velocities of interstitial fluid and formalin [46]. As formalin diffuses into a tissue specimen and replaces exchangeable fluid (e.g. interstitial fluids), the overall composition of the tissue is physically altered resulting in a TOF differential. As faster formalin diffuses into the sample the tissue's net sound velocity increases resulting in a monotonic decrease to the TOF signal. However, in our previous work the functional relationship between TOF-based diffusion signals and stain quality remained elusive. In this work, we report the development of a real-time statistical model and custom-modified tissue processor system that determines when a sample is adequately diffused enough to produce high-quality staining from downstream IHC assays.

## Methods

### Tissue acquisition and fixation

Human tonsil tissue was obtained fresh and unfixed from a local Tucson, Arizona hospital under an approved contractual agreement for procurement of biological samples. In accordance with approved protocol, all tissue samples were completely de-identified, if applicable, and anonymized so there was no possibility that anyone could link the specimen back to a patient. Furthermore, in accordance with protocol, all tissue samples were collected under the requirement that the tissue was medical waste material (i.e. leftover diagnostic specimens) such that no informed consent was required. All data was analyzed anonymously. Because all biological specimen were fully anonymized tissue that was medical waste material, no internal ethics review committee or IRB was required for these studies. Whole tonsils from same day surgeries were transported to Roche Tissue Diagnostics on wet ice in biohazard bags. Samples of tonsil tissues of precise sizes were obtained by using 6 mm diameter biopsy punch (Miltex #33–36). For fixation experiments from a single organ, four 6 mm cores were obtained from the same whole unfixed tonsil organ. One core was used as a positive fixation control and was placed into room temperature NBF for 24 hours (RT, N = 24). One core was used as a positive fixation control for cold+hot fixation, and was placed for 6 hours into 10% NBF (Saturated aqueous formaldehyde from Fisher Scientific, Houston, Texas, buffered to pH 6.8–7.2 with 100 mM phosphate buffer) previously chilled to 4˚. This control core was removed and placed into 45˚C NBF for an additional 1 hour to initiate crosslinking (6+1, N = 24). One core was used as an intentionally underfixed control for cold+hot fixation, and was placed for 2 hours into 10% NBF for previously chilled to 4˚. This underfixed control core was removed and placed into 45˚C NBF for an additional 1 hour to initiate crosslinking (2+1, N = 21). One core was secured in a tissue cassette and placed in between the TOF sensors on the automated fixation device and the algorithm and tissue processor monitored the status of diffusion (Dynamic, N = 18). After fixation, all samples were processed in a commercial tissue processor set to an overnight cycle and embedded into wax.

### Time-of-flight measurement

As previously described, we developed a digital acoustic interferometry algorithm that calculated TOF differentials with subnanosecond precision [46]. Briefly, pairs of 4 MHz focused transducers were spatially aligned and tissue samples were placed at their common foci. One transducer was programmed to send out a sinusoidal pulse that was detected by an accompanying transducer after traversing the formalin and tissue and the received pulse was used to calculate the transit time. An initial calibration TOF reading was acquired by measuring the signal through only formalin. That baseline reading was subtracted from the TOF with the tissue present to isolate the phase alteration from the tissue. This detection method simultaneously isolated the acoustic TOF shift due to diffusion and compensated for

environmentally-induced fluctuations in the formalin. For each tissue, TOF measurements were recorded every 1 mm along the vertical axis of the tissue, with typically 5–9 total measurements per tissue depending on the size of the biospecimen. All of the TOF signals for a given tissue were then averaged together and the spatially-averaged signal was recorded as representative of the tissue's overall rate of formalin diffusion. In practice, the form of the TOF diffusion signal from multiple tissue types was well-correlated with a single-exponential decay function [47].

## Histology

Paraffin tissue blocks were sectioned to 4 μm thick and placed onto Fisherbrand™ Superfrost™ Plus microscope slides (Thermo Fisher Scientific). One section was stained with H&E on a Ventana Medical Systems HE600 automated staining system to evaluate tissue morphology. In addition to the H&E slide, two serial sections were cut and stained with anti-FOXP3 (SP33) and anti-bcl-2 (SP66) with Di-aminobenzidine (DAB) for evaluation of stain intensity and coverage. For each stain, staining was performed according to the manufacturer's protocol on a Ventana Benchmark Ultra XT automated staining system.

## Statistical methods

The statistical model used to calculate in real-time when the TOF signal was representative of the tissue's actual diffusion rate was initially developed using custom developed code written in MATLAB (Mathworks) using multiple functions from the Statistics and Machine Learning toolbox. When the final model was translated to the laboratory for implementation on our TOF-scanning hardware, it was translated into Python using several libraries including numpy, matplotlib, scipy, and kapteyn. To evaluate the data from different fixation methods, a Welch's 2-sided t-test with a cutoff for statistical significance of $p < 0.002$ was used.

## Imaging and image processing

Each slide was imaged on a whole slide scanner (VENTANA iScan HT slide scanner) at 20X magnification with DAB stain and a hematoxylin counterstain. A custom developed software package written in MATLAB was used to analyze the images. A segmentation algorithm differentiated the entire tissue section from the background. A separate algorithm identified regions within the tissue footprint that comprised non-staining tissues (e.g. holes, cracks, stroma), to yield the most representative statistics around percent of tissue staining. The transmission images were log transformed and spectrally unmixed to isolate the DAB staining from the hematoxylin counterstain. The unmixed DAB concentration mappings were used to determine which pixels were DAB positive using a global threshold. Raw DAB concentrations were also recorded so the intensity of DAB staining could be analyzed. To study the effects of improper fixation, an algorithm was written that calculated the Euclidean distance to the nearest edge pixel so that intensity of DAB stain and DAB positivity could be studied for different fixation protocols.

## Results

### Criterion for real-time prediction of stain quality

To construct a model that could predict optimal fixation quality in real-time, we needed to understand the relationship between formalin diffusion and morphological properties. Diffusion is controlled mainly by concentration gradients and time according to Fick's Laws of diffusion. Time course experiments (1.5, 3 and 6 hours) were performed using 6 mm cores of

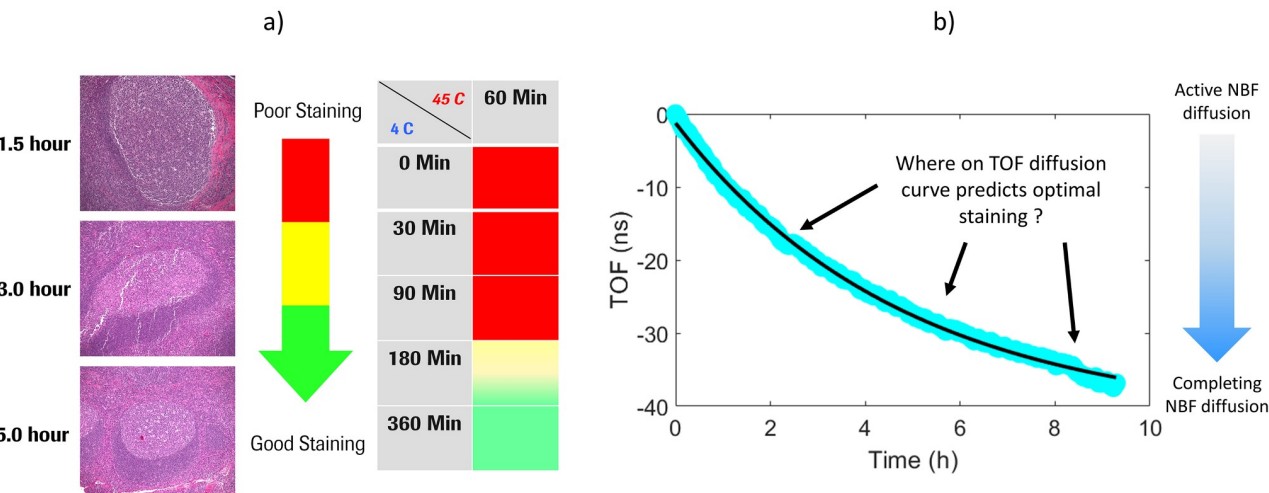

**Fig 1. Correlating NBF diffusion times with stain quality.** a) H&E images acquired with different cold soak times in NBF. Based on H&E based morphology, cold soak times of 1.5 hours, 3 hours, and 5 hours produced severely compromised, borderline, and exemplary staining, respectively. b) Example depiction of TOF diffusion curve with active NBF diffusion immediately after submerging tissue in NBF, manifesting with a rapidly changing TOF signal. Conversely, after several hours the tissue's rate of diffusion had significantly slowed as the tissue and NBF approached osmotic equilibrium.

human tonsil tissues submerged into 4˚C NBF followed by 1 hour in 45˚C NBF [45]. After repeating the time course for 10 tonsil organs a minimum of 3 hours of cold NBF (C/H:3+1) was determined to produce acceptable histomorphology. Tissue morphology was improved with 5 hours in cold NBF (C/H:5+1) but further cold soak times provided no additional benefit. An additional 10 tonsil cores were then examined to confirm that a C/H:5+1 protocol produced high-quality staining, see cumulative results in Fig 1a.

In the previous section, the diffusion times required to produce high-quality H&E staining were empirically determined. Next, we quantitatively characterized the diffusivity properties of human tonsil tissues to develop an analytical metric to determine when a sample was properly diffused. Several whole tonsil tissues were cored to 6 mm in diameter. A total of 38 6 mm tonsil samples were measured in cold (7±0.5˚C) 10% NBF. Of the 38 samples, 14 were monitored for 3 hours and the remaining 24 samples were monitored for 5 hours. For each sample the diffusion was measured throughout the sample (1 mm intervals) and the spatially-averaged TOF curves were calculated. The characteristic TOF-based diffusion signals were highly correlated with a single-exponential curve of the form,

$$TOF(t) = A_{avg} * e^{\frac{-t}{\tau_{avg}}} + C \tag{1}$$

where $C$ is a constant offset in nanoseconds, $A_{avg}$ is the amplitude of the exponential decay in nanoseconds, and $\tau_{avg}$ is the tissue's average decay constant in hours. Samples scanned for 3 hours and 5 hours had average decay constants of 2.33 hours and 2.72 hours, respectively. The difference of 0.39 hours was statistically insignificant, indicating that the two datasets faithfully measured the same physical phenomena. This established that the TOF measurement system produced consistent and reproducible results for long and short diffusion times.

Having validated the diffusion monitoring system, the dataset of 38 6 mm tonsil samples was analyzed to find a correlation between the diffusive properties of each tissue and the empirically determined diffusion times required to produce ideal downstream staining. Numerous analytic techniques were employed including multivariate analysis, cluster based algorithms, characterization of the signal's derivative, and principal component analysis. A

slope based analysis, physically representing the rate of diffusion, was found to provide ideal and meaningful discrimination of samples that were in cold formalin for 3 hours (i.e. adequately staining) versus 5 hours (i.e. ideally staining throughout the sample). To significantly mitigate noise and more accurately represent the active rate of diffusion, the derivative of the TOF signal was calculated based on a fit to a single exponential function. Additionally, ideal discrimination of the two datasets was achieved by amplitude-normalizing each signal,

$$m(t = t_o) = 100 \left( \frac{-1}{\tau_{avg}} e^{\frac{-t_o}{\tau_{avg}}} \right) \left[ \frac{\%}{hr} \right] \tag{2}$$

where $m$ is the derivative of the amplitude-normalized TOF signal at time $t_o$ and the brackets denote the slope's units of percent TOF change per hour of diffusion. Fig 2a displays the normalized slope, at 3 hours and 5 hours respectively, of each sample. The average rate of diffusion at 3 hours was -11.3%/hr, whereas at 5 hours the rate of diffusion had significantly slowed to -5.3%/hr, with several samples approaching full osmotic equilibrium as indicated by a near zero rate of diffusion. The different distributions of normalized rates of diffusion at 3 hours and 5 hours were highly statistically significant ($p < 2e-15$), indicating a drastic and physically real difference in the rate of diffusion at 3 hours versus 5 hours. Similarly, we observed a difference in H&E-based morphology for samples with cold diffusion times of 3 hours and 5 hours.

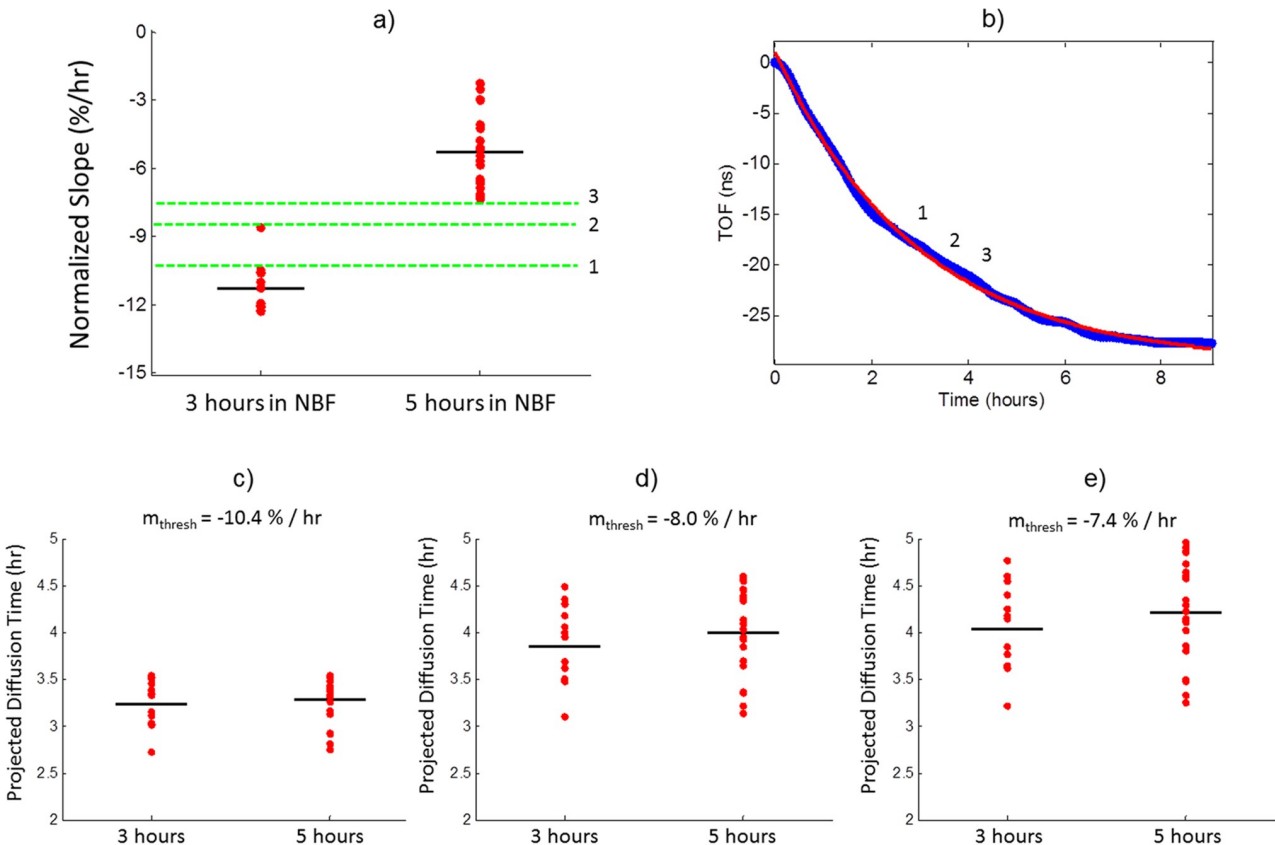

**Fig 2. Tuning rate of diffusion metric to predict stain quality.** a) Normalized slope for tonsil tissues at 3 hours (N = 14) and 5 hours (N = 24) in cold NBF when tissues are expected to have borderline and ideal staining, respectively. Different threshold diffusion rates of -7.4%/hr, -8.0%/hr, and -10.4%/ hr are indicated with 3, 2, and 1 respectively. b) Average TOF signal from a 6 mm tonsil with the approximate locations of the threshold diffusion rates indicated. c-e) Plots of the projected completion times for each of the three evaluated threshold diffusion rates.

We conclude that the TOF-based diffusion metric has sufficient capability and sensitivity to discriminate between poorly fixed (i.e. 3 hours of NBF diffusion) and well-fixed (i.e. 5 hours of NBF diffusion) tissue samples. The correlation between TOF-based diffusion metrics and H&E-based stain quality, indicates our diffusion monitoring system, if properly calibrated, was fundamentally capable of predicting eventual stain quality. Given this validation, Eq. 2 was solved for the time required to reach a criterion rate of NBF diffusion,

$$t_{done}(\tilde{m}) = -\tau_{avg}\ln\left(|\tilde{m}_{thres} \cdot \tau_{avg}|\right), \qquad (3)$$

where $t_{done}$ is the time required to reach a threshold slope ($m_{thresh}$), and the $|...|$ symbol indicates the absolute value. For a given tissue specific decay constant and normalized rate of diffusion, this equation can be used to calculate how long a sample needs to be in cold formalin before it will reach a threshold rate of diffusion. To evaluate rate of diffusion as a stain quality predictor, three threshold slope values ($m_{thres}$ = -7.4%/hr, -8.0%/hr, -10.4%/hr) were chosen for evaluation, as displayed visually in Fig 2a. Note that large absolute slope values represent more fluid exchange per hour. Thus as a sample's active diffusion slows, the rate of diffusion will approach osmotic equilibrium (i.e. 0%/hr). Therefore, a larger threshold slope criterion predicts samples have adequate formaldehyde sooner. This is illustrated graphically in Fig 2b on a representative 6 mm piece of human tonsil, where decreasing diffusion rates translate to longer completion times.

Additionally, the projected completion times for tissues in each dataset (3 hours and 5 hours) are displayed in Fig 2c-2e) as calculated from Eq. 2. For example, Fig 2c displays the projected completion times for the largest threshold slope of -10.4%/hr. This slope criterion predicted an average completion time of 3.27 hours. However, 6 of the sample's (16%) projected completion times are less than 3 hours and from our previous histological staining results, these samples are known to not be adequately diffused throughout. Additionally, one sample would be misidentified with this slope criterion. Based on these findings all samples must not have sufficient formalin to stain acceptably when their rate of diffusion is -10.4%/hr. The middle threshold slope value of -8.0%/hr produces ideal discrimination between the two datasets and reasonable completion times between 3 and 5 hours. However, to be as conservative as possible, the lowest threshold slope value of -7.4%/hr was selected as the criterion for when samples will stain ideally throughout. With this metric, 6 mm tonsil cores were projected to take between 3.21 hours and 4.96 hours, which we know from downstream histological staining to be reasonable. Based on these experiments and analysis, once a sample's real-time rate of normalized diffusion slows to,

$$\tilde{m}_{thres}(t) \leq -7.4\%/hr, \qquad (4)$$

the sample will have sufficient formalin throughout to guarantee ideal and uniform histological staining.

## Statistical model for real-time validation of TOF

**Signal.** A practical embodiment of the TOF technology requires the diffusion curve to be analyzed with temporally sparse data in the absence of a ground truth assessment of the tissue's true temporal diffusion profile. For instance, the prediction of when a sample is optimally fixed is based on the detected rate of NBF diffusion at a given time. However, this prediction is only valid if the exponential fit of the tissue's diffusion curve is representative of the tissue's actual rate of diffusion. In particular at the beginning of an experiment when data is sparse, the detected rate of diffusion can change significantly due to a variety of sources (e.g. tissue

deformation, thermal noise, fluidic variables, etc.) in addition to noise inherent to the TOF cal-culation. To overcome these limitations, we developed a statistical model to validate when the TOF curve had converged to the true diffusion profile of the tissue and was therefore accurate enough to make a prediction as to whether the sample was adequately fixed.

The developed solution was a statistical model that had three independent components that query different components of the TOF signal. All three components were continually re-cal-culated as TOF data points were calculated during cold NBF diffusion. When all three statisti-cal parameters were simultaneously satisfied, the diffusion profile was judged to be accurate.

Condition #1 (Past Algorithm): This condition established that the current TOF fit was consistent with previously collected data. The decay constant of the single exponential line must have converged, as defined by 10 consecutive decay constants having less than a 2% dif-ference relative to the average of the previous 6 decay constants.

Condition #2 (Present Algorithm): This condition established the statistical confidence of the presently collected data. The 95% percent confidence interval of the most recent decay con-stant must be below 0.7 hours.

Condition #3 (Future Algorithm): This condition established that there was sufficient statis-tical confidence to predict future TOF values. The 95% confidence interval of the TOF signal across all times (past, present, and future) must be below 2 ns.

Once all three statistical conditions were satisfied, the algorithm validated that the TOF sig-nal from the tissue was representative of the actual rate of diffusion and predicted at what time the tissue will be properly fixed according to Eq. 2 with the threshold condition described in Eq 3. A graphical depiction of the real-time statistical model is displayed in Fig 3a where the TOF diffusion signal was judged representative at 3.08 hours, at which point the predictive algorithm calculated the tissue would be properly diffused at 4.47 hours. The calculation of the past, present, and future algorithms, up to the point they were satisfied, is plotted in Fig 3b)-3d), respectively. Additionally, a video of the model operating on data in real-time from a 6 mm piece of tonsil tissue is presented as supplemental data (S1 Video).

The overall efficacy of the methodology was tested on 105 previously collected TOF curves [47] from 16 different tissue types (Brain, Breast, Cervix, Colon, Duodenum, Lung, Gallblad-der, Jejunum, Kidney, Liver, Lymph Node, Prostate, Skin, Testis, Tonsil, Uterine). Addition-ally, for breast, colon, liver and testis tissue the model was tested and validated on healthy and cancerous tissue. A simulation was written in which this *a priori* data was analyzed by the sta-tistical model to simulate how it would behave on real-life empirical data. At the time the model validated the signal, the predicted time of optimal fixation was calculated to be within 27 minutes of the true time, as calculated by the fit at the end of the experiment. The statistical confidence of the decay constant at the time of convergence was ±7 minutes. These results confirm that the model determined in real-time when the TOF signal was representative of the tissue's actual rate of diffusion and that predictions based on that data were accurate. Fig 3e) plots the time to validate characteristic diffusion curves versus the predicted fixation times. On average, the model converged to a valid fit 55 minutes before the sample was properly diffused. These results detail how the model was capable of determining when biospecimens will be ide-ally diffused nearly an hour before they were, and thus established the feasibility of a commer-cial embodiment of TOF-based diffusion monitoring coupled with a real-time assessment of fixation quality.

## Validation of fixation predictive model with IHC

**Staining.** To confirm that the developed predictive model was truly predictive of eventual stain quality, we developed digital pathology tools to objectively and repeatably evaluate the

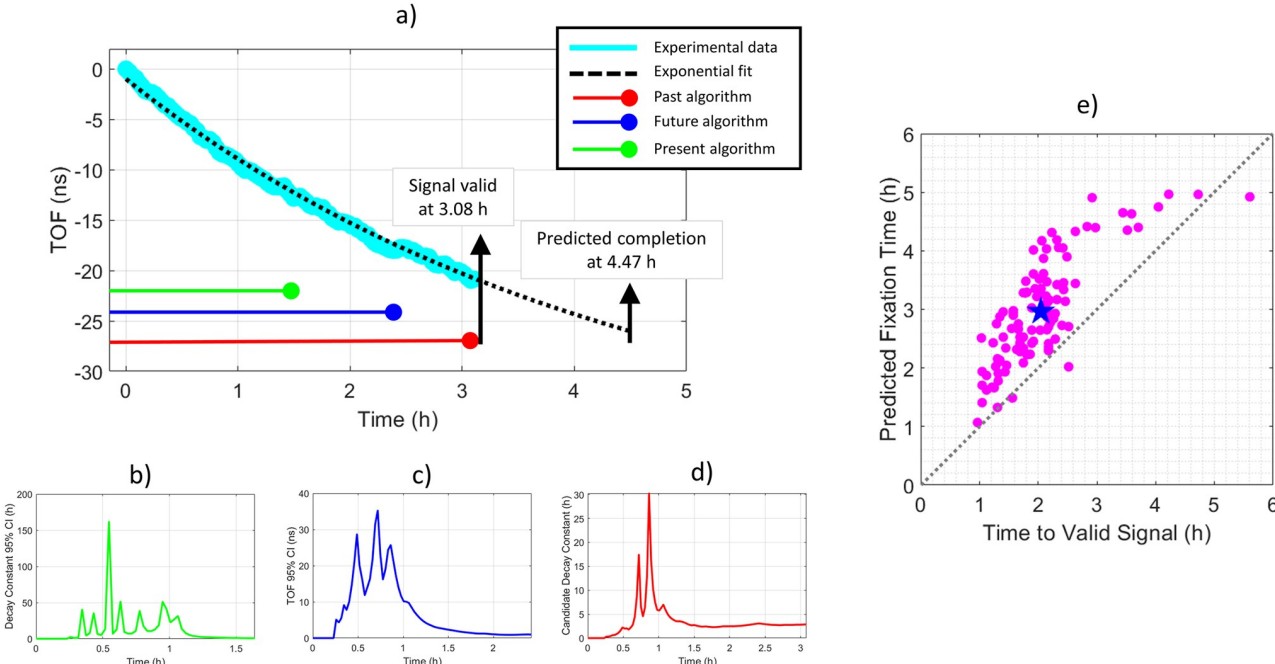

**Fig 3. Example of statistical model to validate TOF diffusion signal.** a) TOF diffusion curve demonstrating how the validation algorithm works during real-time data acquisition. As data is collected, three statistical algorithms continually monitor the fidelity of the signal focusing on past, present, and future aspects of the data. Time to satisfy each algorithm is labeled on the plot with solid horizontal lines in order of convergence time. Once all three conditions of data fidelity are satisfied, the data is representative of the tissue's actual rate of diffusion and the required diffusion time of the tissue was calculated. Detailed views of the present, future, and past algorithms are shown in Fig 4 b), c), and d) respectively. e) Plot of time to validate diffusion signal versus predicted fixation times for 105 tissues from 16 distinct organs. On average (blue star), 2.04 hours was required for the detected diffusion profile to converge onto the tissue's actual rate of diffusion and tissues were predicted to need 2.96 hours of diffusion in cold formalin.

level of staining of each slide. Digital analysis was performed on IHC slides with DAB staining a single marker and hematoxylin counterstain. A whole slide scan of each slide was acquired. The software segmented the tissue, and identified and removed areas that did not stain, such as connective tissue and stroma. For all regions with active staining, the software quantified the percent of the tissue that was DAB positive. Additionally, it calculated the edge distance, defined as the shortest possible distance to the border of the tissue sample. Next radial concentric 0.33 mm "zones" of the tissue were calculated so staining could be analyzed in regions with roughly equal concentrations of formalin. This geometrical representation of the tissue enabled the effects of improper fixation to be explicitly analyzed because formalin diffuses into a tissue from the periphery resulting in progressively less staining at the core of the tissue. A histogram was calculated plotting the percent of the tissue staining versus distance to the nearest tissue edge. Finally, we analyzed each radial zone of the tissue for proper staining by defining suppressed staining as DAB positivity less than half of the edge/maximum positivity. A graphical depiction of the image analysis workflow is presented in Fig 4.

To confirm that our developed metrology could accurately ascertain in real-time when a tissue was optimally diffused, we performed a large scale study with 87 distinct tonsil cores differentially fixed with one of four fixation protocols. For all fixation protocols, after fixation tissues were processed in a standardized fashion, embedded in paraffin blocks, and slides were cut and stained with DAB for FOXP3 and bcl-2 (see supplemental data). The first three fixation protocols were: cold NBF for 2 hours followed by heated NBF for 1 hour (C/H: 2+1, N = 24), cold NBF for 6 hours followed by heated NBF for 1 hour (C/H: 6+1, N = 21), and

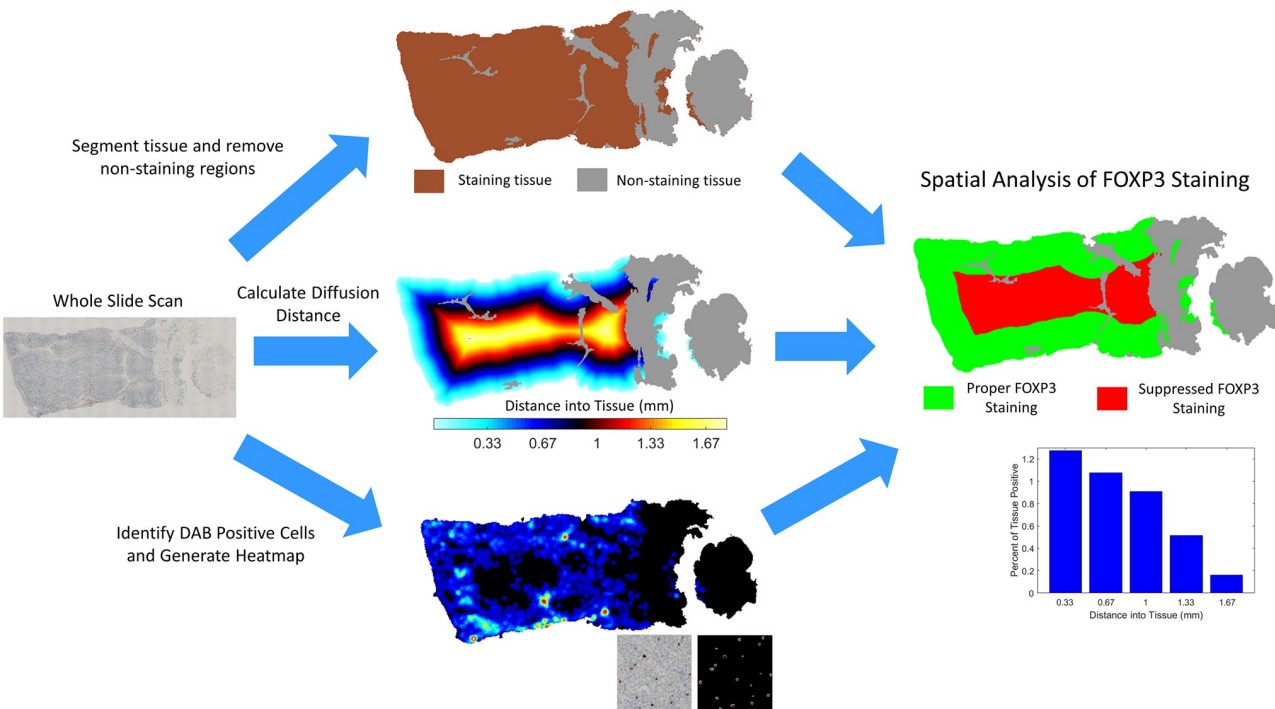

**Fig 4. Schematic of radial image analysis workflow used to analyze the impact of formalin diffusion on functional IHC staining to FOXP3.**

room temperature NBF fixation for 24 hours (RT:24hr, N = 24). Finally, we tested our real-time fixation prediction algorithm using the cold+hot fixation method in which tonsils were placed in cold NBF until our system determined they were sufficiently diffused at which point they were moved to heated formalin for 1 hour to initiate crosslinking (C/H:Dynamic, N = 18). In this experiment, the C/H:6+1 and RT:24hr protocols represented internal controls because these fixation methods were known to produce high-quality staining. Alternatively, the C/H:2+1 samples represented intentionally underfixed tissues. Representative images of the staining patterns for each fixation method are displayed in Fig 5. The two standard fixation protocols produced tissue that stained nearly uniformly whereas the C/H:2+1 tissue had significantly suppressed staining at the center of the tissue. Importantly, the tissue from the C/H: Dynamic protocol produced uniform FOXP3 staining consistent with the two gold-standard fixation protocols.

Furthermore, cumulative box and whisker plots for the entire 87 tissue study for FOXP3 staining are presented in Fig 6. The stain penetration depth, as defined by the distance at which staining drops to half its edge value, is plotted in Fig 6a and the area of the tissue that properly stained is plotted in Fig 6b. For both metrics the C/H:2+1 samples exhibited significantly reduced staining compared to the other three fixation methods ($p<0.002$). Importantly, C/H:Dynamic samples displayed staining results that were concordant with both the RT:24hr samples as well as C/H:6+1 samples. Additionally, dynamically fixed samples were in cold NBF for only 4.2 hours, meaning the TOF predictive algorithm produced tissues that stained as well as traditionally fixed tissues but several hours faster. Additionally, the average spatial stain profile for FOXP3 for all four fixation protocols is plotted in Fig 6c). The dynamically fixed samples again displayed a staining pattern consistent with C/H:6+1 and RT:24hr fixation.

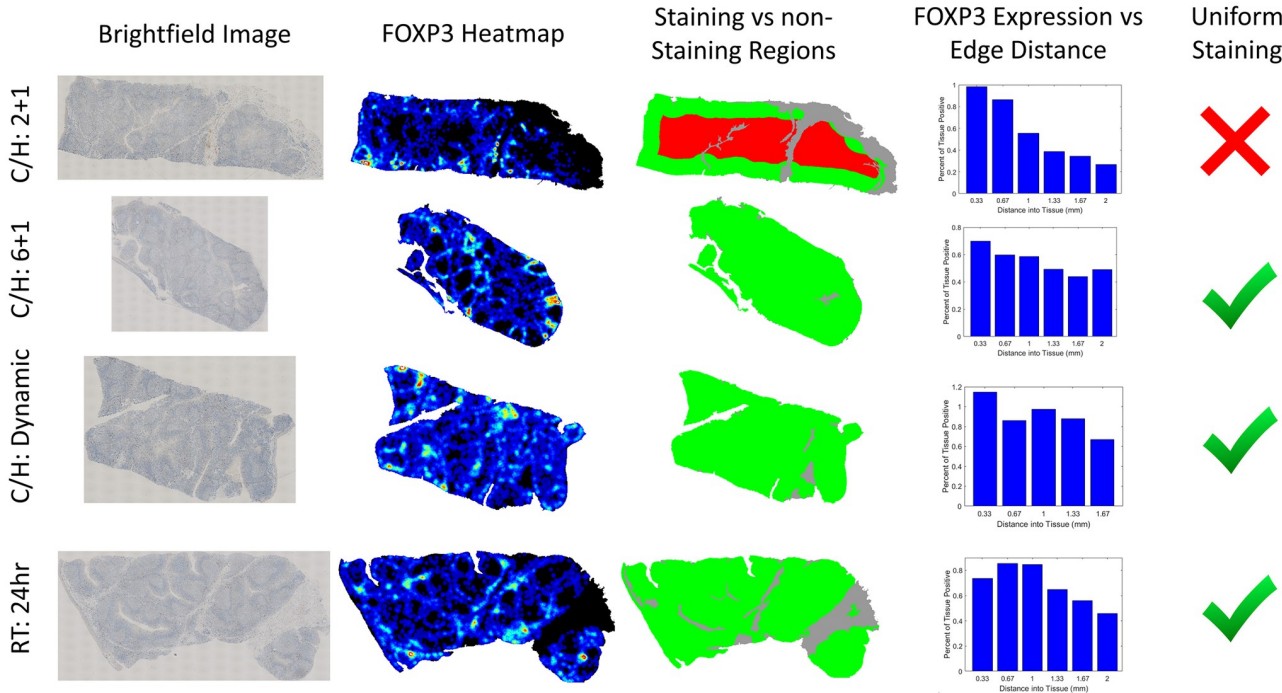

**Fig 5. Example staining results for different fixation protocols.** Left column) Brightfield whole slide scan of tissue. Middle-left column) Heatmap of FOXP3 positivity for each tissue. Middle-right column) Graphical depiction of radial zones within each tissue indicating regions of deficient and proper staining, as defined by DAB positivity within 50% of the edge value. Right column) Histogram of FOXP3 staining versus distance to the edge of the tissue.

This is in marked contrast to the C/H:2+1 samples that had roughly 70% less staining at the center versus the edge of the tissue.

## Discussion

A dominant source of error in the histology laboratory is caused by improperly fixed tissues, resulting in decreased stain intensity and poor morphology [29–31]. In this work, we have shown the development of a first-of-its-kind tissue fixation system capable of determining in real-time when a biospecimen has adequate formaldehyde to guarantee high-quality fixation as demonstrated by ideal and spatially uniform functional staining from downstream IHC assays. The predictive algorithm was trained on a combination of H&E-based morphology as well as characteristic diffusion curves from tonsil samples. Ultimately, our system was validated by comparing staining results from different fixation protocols. Dynamically fixed samples, with diffusion times actively controlled by our statistical model, were found to have equivalent FOXP3 and bcl-2 staining compared to the current clinical gold-standard of 24 hour room temperature fixation.

In today's histology lab, the process of tissue fixation is largely dictated by workflow considerations. At a number of facilities, the timing of starting the tissue processor at the end of the day, to run a standard overnight protocol for the morning shift, dictate fixation times. As a result, fixation times can vary significantly in the same and certainly at different institutions. One difficulty of having varied preanalytic parameters is the inability to share data and results easily across multiple sites [2]. Clinicians would benefit greatly from data collected from standardized samples that were treated with the same fixation protocol. For instance, digital

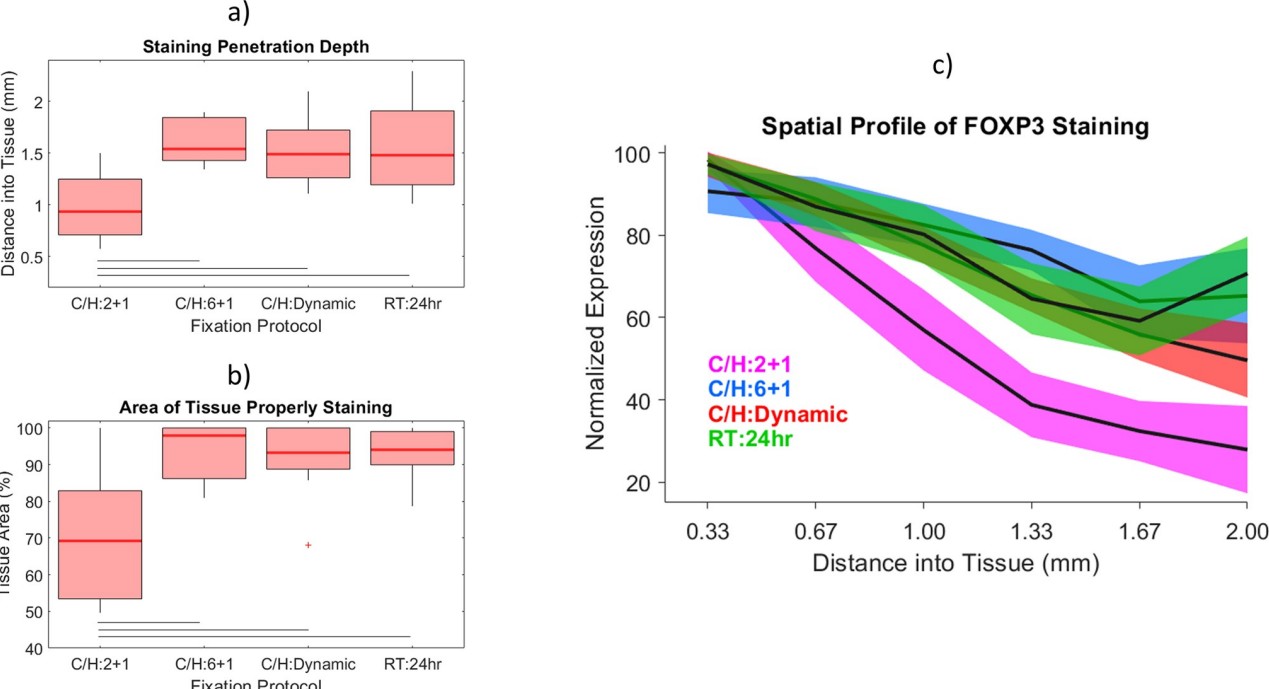

**Fig 6. Quantitative analysis of FOXP3 staining for different fixation protocols.** a) Penetration depth of proper FOXP3 staining, as defined by distance into tissue where staining drops to half its edge value. b) Percent of tissue displaying proper FOXP3 staining. c) Normalized FOXP3 expression plotted versus distance into tissue. Solid lines indicate *p<0.002* according to Welch's 2-sided t-test.

pathology algorithms can output inconsistent results when analyzing samples with improper preanalytical processing. We report more consistent staining levels with properly fixed tissues compared to poorly fixed samples for FOXP3 and bcl-2 staining (S2 Fig). Another benefit of standardizing tissue fixation is increased efficiency which would reduce the amount of costly re-work due to poor quality or lost tissues. The costs of this re-work often falls on the testing lab and could be greatly reduced with a standardized system as part of a fully documented pre-analytical workflow.

We have previously reported that a novel fixation protocol, based on two temperature zones, better preserves labile biomarkers such as phospho-proteins and mRNA [48]. More rapid and efficient techniques, such as two-temperature fixation, will become necessary as the number of samples continues to increase and diagnostic assays rely on next generation bio-markers. We have described here a system to take advantage of this more efficient method of fixing tissues by standardizing formaldehyde penetration. Histology practices have been around for decades and these practices are difficult to change. The methods and instrumentation we describe take advantage of the same reagents (10% NBF) and procedures (formaldehyde crosslinking) and therefore do not alter downstream assay protocols or results. Rather, this and other more efficient methods of tissue standardization will be necessary to enable accurate preservation and quantitation of future analytes.

In this work we have tested and validated our system using FOXP3 expression in human tonsil samples, however, we believe the system will be applicable to all biomarkers and tissue types. For example, additional data relating to faithful preservation of bcl-2 with our novel fixation method is presented in Supplemental material. The dynamic fixation protocol demonstrated equivalent staining to clinical 24 hour room temperature fixation (compare S1 vs S2

Figs). The expression pattern for bcl-2 did not show a heavy dependence on fixation quality, which was expected because it is a known as a robust clinical biomarker. This result highlights that improved fixation will be particularly beneficial for labile biomarkers. Analysis of staining intensity for differentially fixed tissues is presented in S3 Fig. Additionally, we have previously generated diffusion data for 34 different tissue types and shown that each has its own characteristic diffusion rate and therefore our predictive statistical model can be applied to multiple tissue types (S3 Fig and S1 Table) [46].

This work presents a method of optimizing the infiltration of formalin into a tissue specimen. Although fixation quality is a dominant factor to determine tissue quality, multiple additional factors can influence the quality of a sample such as warm and cold ischemia time, fixation time and type, dehydration, paraffin embedding, and storage conditions. One of the next steps for this technology is to explore the use of this predictive algorithm empirically on other tissue types and biomarkers, in particular biomarkers known to be preanalytically sensitive. Future work will entail extending our current criterion, developed with tonsil tissue, to a plurality of tissue types so that a universal fixation criterion can be realized. We postulate that the current metrology can be tuned to create a generalizable model that can optimally predict when all types of tissue are appropriately fixed. We believe the histology lab of the future could employ this analytical method as part of a random-access automated processing unit that could ensure and document that individual tissue samples are optimally fixed, and truly transform tissue fixation into a science.

## Supporting information

**S1 Fig. Quantitative bcl-2 expression for different fixation protocols.** a) Penetration depth of proper bcl-2 staining. b) Area of proper bcl-2 staining. c) Normalized bcl-2 expression plotted versus distance into the tissue.
(TIF)

**S2 Fig. Quantitative analysis of stain intensity for different fixation protocols.** a) Normalized intensity of FOXP3 staining plotted versus distance into the tissue. b) Normalized intensity of bcl-2 staining plotted versus distance into the tissue.
(TIF)

**S3 Fig. Projected optimal fixation times for numerous tissue types based on the developed predictive algorithm.**
(TIF)

**S1 Table. Average decay constant and average predicted optimal fixation times for different tissue types.**
(XLSX)

**S1 Video. Real-time example of algorithm to predict optimal fixation time.** Acquisition of TOF-based diffusion curve from a 6 mm tonsil in 4% NBF with a graphical display of the developed optimal fixation prediction algorithm.
(AVI)

## Author Contributions

**Conceptualization:** Daniel R. Bauer, David R. Chafin.

**Data curation:** Daniel R. Bauer, Torsten Leibold.

**Formal analysis:** Daniel R. Bauer.

**Investigation:** Torsten Leibold, David R. Chafin.

**Methodology:** Daniel R. Bauer, David R. Chafin.

**Project administration:** Torsten Leibold, David R. Chafin.

**Software:** Daniel R. Bauer, Torsten Leibold.

**Supervision:** Torsten Leibold, David R. Chafin.

**Writing – original draft:** Daniel R. Bauer, David R. Chafin.

**Writing – review & editing:** Daniel R. Bauer, David R. Chafin.

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
