## [Decision Letter · Decision Letter 0]

13 Jul 2021

PONE-D-21-16557

Making a Science Out of Preanalytics: An Analytical Method to Determine Optimal Tissue Fixation in Real-time

PLOS ONE

Dear Dr.  Bauer,

Thank you for submitting your manuscript to PLOS ONE. After careful consideration, we feel that it has merit but does not fully meet PLOS ONE’s publication criteria as it currently stands. Therefore, we invite you to submit a revised version of the manuscript that addresses the points raised during the review process.

I read with attention the referees' comments, and I think that the paper would benefit if you could test tissues of different organs and samples obtained using different sampling procedures. From the title a reader would expect molecular analyses, thus at least the analysis of DNA quality should be integrated.

Please submit your revised manuscript by July 30th If you will need more time than this to complete your revisions, please reply to this message or contact the journal office at plosone@plos.org. Please include the following items when submitting your revised manuscript:

We look forward to receiving your revised manuscript.

Kind regards,

Anna Sapino

Academic Editor

PLOS ONE

Journal Requirements:

2. Thank you for stating in the manuscript Methods section: 'Human tonsil tissue was obtained fresh and unfixed from a local Tucson, Arizona hospital under a contractual agreement with approved protocols.'

a. Please amend your current ethics statement to include the full name of the ethics committee/institutional review board(s) that approved your specific study and confirm that your named institutional review board or ethics committee specifically approved this study. 

"DB, TL and DC were full-time employees of Roche Tissue Diagnostics during this work. There are no products in development or marketed products to declare. Roche Tissue Diagnostics has filed patents on the TOF technology and associated algorithms." 

We note that you received funding from a commercial source: Roche Tissue Diagnostics

4. We note that you have stated that you will provide repository information for your data at acceptance. Should your manuscript be accepted for publication, we will hold it until you provide the relevant accession numbers or DOIs necessary to access your data. If you wish to make changes to your Data Availability statement, please describe these changes in your cover letter and we will update your Data Availability statement to reflect the information you provide

6. We note that Figure(s) 4 and 5 in your submission contain map images which may be copyrighted. All PLOS content is published under the Creative Commons Attribution License (CC BY 4.0), which means that the manuscript, images, and Supporting Information files will be freely available online, and any third party is permitted to access, download, copy, distribute, and use these materials in any way, even commercially, with proper attribution. For these reasons, we cannot publish previously copyrighted maps or satellite images created using proprietary data, such as Google software (Google Maps, Street View, and Earth). For more information, see our copyright guidelines: http://journals.plos.org/plosone/s/licenses-and-copyright.

Reviewers' comments:

Reviewer's Responses to Questions

**Comments to the Author**

1. Is the manuscript technically sound, and do the data support the conclusions?

Reviewer #1: Yes

Reviewer #2: Yes

2. Has the statistical analysis been performed appropriately and rigorously? 

Reviewer #1: Yes

Reviewer #2: Yes

3. Have the authors made all data underlying the findings in their manuscript fully available?

Reviewer #1: No

Reviewer #2: Yes

4. Is the manuscript presented in an intelligible fashion and written in standard English?

Reviewer #1: Yes

Reviewer #2: Yes

5. Review Comments to the Author

Reviewer #1: The manuscript presents a tool based on a statistical model to determine the quality of fixation. Although the topic is interesting I believe that the manuscript is difficult to read in the present form and some mandatory minor revisions are needed before its publication.

My concerns:

First of all the revision process is quite challenging without page and row numbers. Please, in the revised manuscript include them.

Introduction: “The most prevalent fixative is 10% neutral buffered formalin (NBF) which is an aqueous solution of formaldehyde in a buffer and has been used for over a century (Fox et al. 1985). Currently, proper fixation protocols are empirically determined by examining the histologically stained tissue for proper morphological features. The result is a mixed bag of adequate and poor morphology depending on the operator, institution, tissue type, and biomarker of interest.” In the last years tissue processing has been ameliorated by the publication of ISO standards for Specifications for pre-examination processes for formalin- fixed and paraffin-embedded (FFPE) tissue. There are three documents at present : specifically for DNA , RNA, proteins. In these it is clearly reported that although the formalin fixation duration can vary depending on the tissue type and size, for tissues pieces of 5 mm thickness fixation duration of 12-24 hours are reasonable for proper penetration and fixation. I suggest including that info in the introduction and commenting standards in the discussion.

Material and methods:

The study is based mostly on previous studies, therefore it is sometime difficult to follow. I suggest including in a supplementary file the relevant data used in this study and derived from previous ones.

Tissue acquisition and fixation: please, define the number of punches processed with the different fixation protocols. “Other punches” or “additionally biopsy punches” should be defined with numbers.

Time of Flight measurement: “Multiple TOF measurements were recorded throughout each tissue specimen and the spatially… Please, define how many measurements were recorded to produce the averaged signal.

Histology: “serial sections of each block” please define the number of sections.

Please, in a supplementary file, please provide the statistical tests used in the study.

Results:

-“Several time course experiments were performed using 6 mm cores of human tonsil tissues submerged into 4°C NBF followed by 1 hour in 45°C NBF (Lerch et al. 2017). After multiple experiments were analyzed, a minimum of 3 hours of cold NBF (C/H:3+1) was determined to produce acceptable histomorphology. Tissue morphology was improved with 5 hours in cold NBF (C/H:5+1) but further cold soak times provided no additional benefit. Multiple cores were then examined and it was confirmed that a C/H:5+1 protocol produced high-quality staining, see cumulative results in Fig 1a. “ Please provide how many cores were examined using the different fixation protocols. The abstract include some numbers that cannot be retrieved in the manuscript. A table could be useful to summarize it.

-Please, provide the number of samples monitored for 3 hours and 5 hours for determining the average slope.

-“Thus, the TOF-based diffusion metric and H&E-based stain quality were highly-correlated, indicating our diffusion monitoring system, if properly calibrated, was fundamentally capable of predicting eventual stain quality.” I cannot find any correlation, but the figure. Do you mean that a visual association between H&E and diffusion rate was made? Please, explain.

Discussion:

“In today’s histology lab, the process of tissue fixation is largely dictated by workflow considerations, rather than on scientific principles, with protocols that can vary significantly at different institutions.” Please, provide a reference for that sentence. I don’t believe that tissue fixation vary significantly. In my opinion the problem is not the mere fixation, but the tissue transport. The latter if made in formalin can have a detrimental effect in case of large surgical specimens.

“One difficulty of having multiple non-standardized procedures is the inability to share data and results easily across multiple sites.” Again, there is the need to define in the discussion “multiple non standardized procedures”.

Please, define the limitations of the study taking into account that tissue processing includes also dehydration steps and paraffin embedding. Improper dehydration can impact on tissue quality, especially in the storage. Furthermore, as limitations the entire study has been developed using tonsil tissues, although from the previous study the diffusion was available also for other tissues (supplementary figure).

Reviewer #2: The manuscript by Bauer and colleagues represents an interesting method for real-time evaluation of the tissue fixation process, using a digital acoustic interferometry algorithm that calculated the TOF differential followed by IHC digital pathology analysis on two markers. The work was performed on a single type of organ (tonsil) for a total of 87 samples in 4 different fixation methods. The analysis is based on different fixation times and temperatures, with deliberate hypo-fixation in some cases.

The real-time monitoring process allows to identify the moment in which the formalin penetration will allow to obtain the best morphological preservation result of the sample.

Both the technical and mathematical method are extremely accurate, as is the written and visual way of disseminating the results.

The English form is excellent, the statistics that are conducted in a rigorous manner.

The main problem is the structure of the experimental design, which consists of some important bugs.

1) Cases were collected from a single type of tissue. Since different organs have different characteristics and fixation times, it would be appropriate to characterize other tissues. Similarly, the type of sampling (biopsy, surgical specimen, cytology) also has different fixation characteristics. Authors should consider completing the study with a pilot project on other tumor types and sampling

2) The analyzed markers are not classical membrane proteins. In order to understand if all cell compartments can benefit from this real-time fission assessment approach, it is recommended to try other markers.

3) Molecular analysis, and therefore the use of nucleic acids in pathological units, has become routine. An evaluation involving tissue fixation for FFPE samples cannot also evaluate the quality of the DNA. It is advisable to extract and qualify at least part of the cases evaluated using, for example, the bioanalyzer

4) The style of both the manuscript and the bibliography is completely wrong, and it does not follow the style of the journal. Please change and correct.

6. PLOS authors have the option to publish the peer review history of their article (what does this mean?). If published, this will include your full peer review and any attached files.

Reviewer #1: No

Reviewer #2: No

---

## [Author Response · Author response to Decision Letter 0]

26 Aug 2021

Editor

I read with attention the referees' comments, and I think that the paper would benefit if you could test tissues of different organs and samples obtained using different sampling procedures. From the title a reader would expect molecular analyses, thus at least the analysis of DNA quality should be integrated.

The authors completely agree with the editor and reviewers that extending the TOF monitoring technology to different types of tissues is a meaningful extension of this work that would be critical to advancing the metrology toward clinical use. However, we have limited ability to address the editor's concern raised around testing our TOF semi-automated diffusion monitoring system with additional types of tissues as these experiments would take several years and resources beyond our current scope. Additionally, due to the COVID-19 pandemic we now have limited ability to engage academic research partners that would have access to fresh unfixed tissues from a variety of different organs. The authors would like to state that although real-time prediction of fixation was only empirically performed on tonsil tissue, we have demonstrated that the method is extendable to multiple other organs as documented in the modeling work done to optimize the predictive algorithm (Figure 3e, additional details added to manuscript) as well as by characterizing the diffusive properties of 34 organs (Supplemental Figure 3) with our TOF technology. Please see response to reviewers for additional details. We have also included this opinion in the discussion section as next critical steps and to perform this comprehensive set of studies with scientific rigor, would require significant resourcing and time. While we will look to extend the technology to other tissue types in the future, we present this work as a first critical step in demonstrating such a predictive algorithm to have substantial interest to the scientific community. Our goal is to present an exciting semi-automated system for real-time diffusion monitoring that would be possible to optimize tissue fixation for histological purposes. To clarify the scope of the project, we would be happy to more clearly define the scope of the work either in the manuscript’s text or in the title itself, or both if need be.

The editor’s and reviewer’s concern about DNA integrity of our C/H fixed tissues has been addressed in the response to the reviewer's where we document extensive unpublished data about this topic. To that point, we have internal and unpublished data that demonstrates equivalent RNA and DNA preservation with cold+hot fixation. This is not a surprising result since the fixation uses standard chemicals (10% Neutral Buffered Formalin) and standard tissue processing techniques. Nonetheless, we have previously checked for DNA integrity and mutation recognition using real time PCR.

Reviewer #1: The manuscript presents a tool based on a statistical model to determine the quality of fixation. Although the topic is interesting I believe that the manuscript is difficult to read in the present form and some mandatory minor revisions are needed before its publication.

My concerns:

First of all the revision process is quite challenging without page and row numbers. Please, in the revised manuscript include them.

Page numbers have been added to all the pages in the manuscript. We did not add row numbers as it can alter the formatting of the document and we are not aware of a robust method of adding row numbers to a Word document. 

Introduction: “The most prevalent fixative is 10% neutral buffered formalin (NBF) which is an aqueous solution of formaldehyde in a buffer and has been used for over a century (Fox et al. 1985). Currently, proper fixation protocols are empirically determined by examining the histologically stained tissue for proper morphological features. The result is a mixed bag of adequate and poor morphology depending on the operator, institution, tissue type, and biomarker of interest.” In the last years tissue processing has been ameliorated by the publication of ISO standards for Specifications for pre-examination processes for formalin- fixed and paraffin-embedded (FFPE) tissue. There are three documents at present : specifically for DNA , RNA, proteins. In these it is clearly reported that although the formalin fixation duration can vary depending on the tissue type and size, for tissues pieces of 5 mm thickness fixation duration of 12-24 hours are reasonable for proper penetration and fixation. I suggest including that info in the introduction and commenting standards in the discussion.

Thank you for pointing out the standards that exist in the anatomical pathology lab. A more relevant reference has been inserted into the Introduction indicating standards for RT fixation (Practical Guide to Specimen Handling in Surgical Pathology, CAP 2020). FYI: The standards that the reviewer raises are for tissues fixed in room temperature formalin and with that protocol are absolutely correct. We have studied formalin fixation extensively and found that the bulk of the time necessary at room temperature fixation is for the crosslinking reaction. Diffusion takes place quickly but crosslinking is kinetically unfavorable and requires significant additional time. Based on studying fixation kinetics (Diffusion + Crosslinking), we use a cold procedure where the diffusion takes place rapidly without crosslinking. Then by raising the temperature, we can drive a very rapid crosslinking reaction with less spatial variability for fixation and have found superior biomarker preservation. We have previously published this Chafin D, et al. (2013) Rapid Two-Temperature Formalin Fixation. PLoS ONE 8(1): e54138. doi:10.1371/

journal.pone.0054138

Material and methods:

The study is based mostly on previous studies, therefore it is sometime difficult to follow. I suggest including in a supplementary file the relevant data used in this study and derived from previous ones.

Thank you for this suggestion, we have now included a supplemental file with the average decay constants (as calculated from previous studies) and the average predicted optimal fixation time for each organ (from the present work)

Tissue acquisition and fixation: please, define the number of punches processed with the different fixation protocols. “Other punches” or “additionally biopsy punches” should be defined with numbers. Thank you for this suggestion, we appreciate the opportunity to clarify our experiments and have revised the text to indicate how many 6 mm Punches were utilized for each tonsil organ. And clarified how those cores were utilized.

Time of Flight measurement: “Multiple TOF measurements were recorded throughout each tissue specimen and the spatially… Please, define how many measurements were recorded to produce the averaged signal. Thank you for pointing this omission out, we have added additional details to the “Time-of-Flight Measurement” section.

Histology: “serial sections of each block” please define the number of sections. Thank you for pointing this omission out, we have specified in the text that, in additional to the H&E slide, two serial sections were cut (one for FOXP3 staining and one for BCL2).

Please, in a supplementary file, please provide the statistical tests used in the study. The statistical test used in this study is a Welch's 2-sided t-test with a cutoff for statistical significance of p<0.002. This has been noted in the caption for Figure 5 as well as the Methods section under the heading “Statistical Methods”.

Results:

-“Several time course experiments were performed using 6 mm cores of human tonsil tissues submerged into 4°C NBF followed by 1 hour in 45°C NBF (Lerch et al. 2017). After multiple experiments were analyzed, a minimum of 3 hours of cold NBF (C/H:3+1) was determined to produce acceptable histomorphology. Tissue morphology was improved with 5 hours in cold NBF (C/H:5+1) but further cold soak times provided no additional benefit. Multiple cores were then examined and it was confirmed that a C/H:5+1 protocol produced high-quality staining, see cumulative results in Fig 1a. “ Please provide how many cores were examined using the different fixation protocols. The abstract includes some numbers that cannot be retrieved in the manuscript. A table could be useful to summarize it.

Thank you for this suggestion, we appreciate the opportunity to clarify how many tonsil cores were utilized in our experiments to examine tissue fixation quality and have revised the text to indicate how many 6 mm Punches were utilized.

-Please, provide the number of samples monitored for 3 hours and 5 hours for determining the average slope.

This data was provided on page 13 above Equation 1. “A total of 38 6 mm tonsil samples were measured in cold (7±0.5 ᵒC) 10% NBF. Of the 38 samples, 14 were monitored for 3 hours and the remaining 24 samples were monitored for 5 hours.” We have highlighted it in the text. Additionally, as these are important details, we have added in text to the Figure 2 legend stating the number of samples used for each respective case. 

-“Thus, the TOF-based diffusion metric and H&E-based stain quality were highly-correlated, indicating our diffusion monitoring system, if properly calibrated, was fundamentally capable of predicting eventual stain quality.” I cannot find any correlation, but the figure. Do you mean that a visual association between H&E and diffusion rate was made? Please, explain.

Our explanation of the data is unclear and we thank the reviewer for pointing this out. We observed a difference in morphology on H&E stained slides that were poorly fixed with 3 hours of NBF diffusion time compared to those that were properly fixed with 5 hours of cold diffusion time. Similarly, the average normalized TOF slope after 3 hours and 5 hours of cold diffusion was statistically different. From this data we conclude that that the TOF-based diffusion metric has sufficient capability and sensitivity to discriminate between tissue samples that are poorly fixed and well fixed. We have added clarifying language to this paragraph to better explain these results and our reasoning. 

Discussion:

“In today’s histology lab, the process of tissue fixation is largely dictated by workflow considerations, rather than on scientific principles, with protocols that can vary significantly at different institutions.” Please, provide a reference for that sentence. I don’t believe that tissue fixation vary significantly. In my opinion the problem is not the mere fixation, but the tissue transport. The latter if made in formalin can have a detrimental effect in case of large surgical specimens. Thank you for the opportunity to clarify the language. We have added additional references in the text as supporting evidence. FYI: In a decade of studying careful tissue collection practices, the suggestion that transport is the significant cause of variability is simplifying the situation in our humble opinion. After visiting multiple medical institutions and sites (large and small hospitals, reference labs and academic hospitals) it is clear there are multiple causes of variability. The two biggest factors that we have found are cold ischemia (the time the sample sits at RT after surgery) and under fixation caused by workflow considerations (ie. late surgeries leading to inadequate time in NBF and needing to start the processor in time for the morning shift). The former is a leading cause of biomarker change (both loss and addition) and the latter a leading cause in signal variability, mostly under fixation causing staining loss. In addition to these two significant variables, we and others have acknowledged differences in tissue handling procedures from different sites and is well known in the field as a significant challenge to overcome for the next generation of high profile biomarkers.

“One difficulty of having multiple non-standardized procedures is the inability to share data and results easily across multiple sites.” Again, there is the need to define in the discussion “multiple non standardized procedures”. Thank you for the opportunity to clarify the language. We have revised the text to include clarifying text and references.

Please, define the limitations of the study taking into account that tissue processing includes also dehydration steps and paraffin embedding. Improper dehydration can impact on tissue quality, especially in the storage. Furthermore, as limitations the entire study has been developed using tonsil tissues, although from the previous study the diffusion was available also for other tissues (supplementary figure). 

As the review points out, there are limitations of the present study. We have added language to the discussion (page 27) to address this issue. While we have extensive experience with the Cold+hot fixation method and biomarker staining in many tissue types, we agree extending the TOF monitoring technology to additional types of tissue is a meaningful and logical extension, however due to the COVID-19 pandemic we have limited ability to engage academic research partners that would have access to fresh unfixed tissues from a variety of different organs. In addition to supplemental Figure 3 the reviewer referenced, the authors would like to highlight that the method is extendable to multiple other organs as documented in the modeling work done to optimize the predictive algorithm (Figure 3e). This modeling work was performed using characteristic TOF curves from 16 distinct organs types to provide a robust demonstration that the real-time prediction algorithm works on multiple types of tissues. We have added text to the caption for Figure 3 to note the number of organs used for the statistical modeling. We have added a list documenting all 16 types of tissues into the text of the manuscript (Page 20). Furthermore, for four of the organs the model was validated on healthy as well as cancerous tissue (Breast, Liver, Colon, Testicular). We have also added this detail into the manuscript on page 20. While we agree extending the technology in the future, we hope that the current work demonstrating a first of its kind capability is publishable in its current form.

Reviewer #2: The manuscript by Bauer and colleagues represents an interesting method for real-time evaluation of the tissue fixation process, using a digital acoustic interferometry algorithm that calculated the TOF differential followed by IHC digital pathology analysis on two markers. The work was performed on a single type of organ (tonsil) for a total of 87 samples in 4 different fixation methods. The analysis is based on different fixation times and temperatures, with deliberate hypo-fixation in some cases.

The real-time monitoring process allows to identify the moment in which the formalin penetration will allow to obtain the best morphological preservation result of the sample.

Both the technical and mathematical method are extremely accurate, as is the written and visual way of disseminating the results.

The English form is excellent, the statistics that are conducted in a rigorous manner.

The main problem is the structure of the experimental design, which consists of some important bugs.

1) Cases were collected from a single type of tissue. Since different organs have different characteristics and fixation times, it would be appropriate to characterize other tissues. Similarly, the type of sampling (biopsy, surgical specimen, cytology) also has different fixation characteristics. Authors should consider completing the study with a pilot project on other tumor types and sampling

The authors completely agree that extending the TOF monitoring technology to different types of tissues is a meaningful extension of this work that would be critical to advancing the metrology toward clinical use. Reviewer #1 also brought up the concern that the model was only demonstrated with one tissue type. We have added significant details to the manuscript to detail that the modeling was validated on 16 different types of organs and in 4 situations using healthy as well as cancerous tissue. Please refer to comment to reviewer #1 for additional details. 

2) The analyzed markers are not classical membrane proteins. In order to understand if all cell compartments can benefit from this real-time fission assessment approach, it is recommended to try other markers. Thank you for the insightful comment regarding the use of FoxP3 and bcl-2 in our study. We agree there are only stains for two proteins in this study, we are confident that staining for proteins in general will benefit from this workflow based on significant past data. The TOF monitoring technology detailed here is merely a semi-automated method of performing Cold+Hot (C/H) fixation. In that respect, we have published work since 2013 detailing the impact of C/H on multiple proteins, tissue types and especially phosphoproteins, where the C/H technology has a clear advantage. The use of FoxP3 and bcl-2 are protein biomarkers that are particularly sensitive to inadequate fixation (which was one of the possible outcomes of our experiments). The use of FoxP3 and it’s subsequent staining (punctate cellular staining) allowed us the opportunity to develop digital algorithms to track anomalies that might be hard to pick up by the naked eye. The inclusion of bcl-2 allowed us to visually track anomalies in the staining pattern as well as digitally as this biomarker is in abundance. Finally, it was not our intent to re-publish the many 10’s of biomarkers that we have examined over the past decade but to rather focus on the ability to develop a tool that might allow more efficient and standardized fixation practices (that seems to be well received). Many of the publications are included in the bibliography and throughout the manuscript already for review. 

3) Molecular analysis, and therefore the use of nucleic acids in pathological units, has become routine. An evaluation involving tissue fixation for FFPE samples cannot also evaluate the quality of the DNA. It is advisable to extract and qualify at least part of the cases evaluated using, for example, the bioanalyzer Thank you for this insightful comment regarding the status of nucleic acids in these samples. We have previously extracted nucleic acids from similar samples, derived from our cold+hot fixation method. In that study, we compared 15 samples for nucleic acid extraction (DNA), analyzed with the bioanalyzer for concentration, compared read length analysis for different sequencing coverages and conducted PCR based mutational analysis for the BRAFV600E single nucleotide change. Samples for Cold+hot were compared to standard 24 hour room temperature paired samples (samples were produced from the same original sample). In all cases the results were equivalent for extraction efficiency, read length analysis and the ability to call the single base alteration for BRAFV600E. The data remains unpublished as there is nothing remarkable about the C+W technique that confers an advantage to nucleic acid based analysis. Other than the samples can be fixed in a more efficient manner (much shorter time). It should be noted that a 3rd party provider provided all of the nucleic acid based extractions and sequencing and were blinded to the sample type. Since nucleic acids are not the focus of this manuscript or effort, this data is not published since there was nothing remarkable. These results are not unexpected since the C+W method still employs fixation with 10%NBF (ie. standard crosslinking) and routine tissue processing chemicals. Furthermore, one could imagine alternate workflows where partial samples are taken for different workflows if a particular lab preferred and there would not be the need to subject the entire sample to the workflow suggested here (future workflow considerations). However, as in our other publications, protein biomarker integrity can suffer with RT procedures.

4) The style of both the manuscript and the bibliography is completely wrong, and it does not follow the style of the journal. Please change and correct.

We are sincerely sorry for the oversight of the format of the bibliography and have corrected it to journal standards.

---

## [Editor Report · Decision Letter 1]

29 Sep 2021

Making a Science Out of Preanalytics: An Analytical Method to Determine Optimal Tissue Fixation in Real-time

PONE-D-21-16557R1

Dear Dr. Bauer,

We’re pleased to inform you that your manuscript has been judged scientifically suitable for publication and will be formally accepted for publication once it meets all outstanding technical requirements.

Kind regards,

Anna Sapino

Academic Editor

PLOS ONE
---

## [Editor Report · Acceptance letter]

4 Oct 2021

PONE-D-21-16557R1 

Making a Science Out of Preanalytics: An Analytical Method to Determine Optimal Tissue Fixation in Real-time 

Dear Dr. Bauer:

I'm pleased to inform you that your manuscript has been deemed suitable for publication in PLOS ONE. Congratulations! Your manuscript is now with our production department. 

Kind regards, 

on behalf of

Dr. Anna Sapino 

Academic Editor

PLOS ONE